# Gas Sensitive Materials Based on Polyacrylonitrile Fibers and Nickel Oxide Nanoparticles

Bayan Kaidar [1,2,*], Gaukhar Smagulova [1,2], Aigerim Imash [1,2] and Zulkhair Mansurov [1,2]

1 Institute of Combustion Problems, Almaty 050012, Kazakhstan
2 The Faculty of Chemistry and Chemical Technology, Al-Farabi Kazakh National University, Almaty 050040, Kazakhstan
* Correspondence: kaydar.bayan@gmail.com; Tel.: +7-7088015015

**Abstract:** The results of the synthesis of PAN/NiO composite fibers by the electrospinning method are presented. The electrospinning installation included a rotating drum collector for collecting fibers. Nickel oxide nanoparticles were synthesized by solution combustion synthesis from nickel nitrate and urea. It was shown that monophase NiO nanoparticles with average particle sizes of 154 nm could be synthesized by this method. NiO nanoparticles were investigated by X-ray diffraction analysis and scanning electron microscopy. Based on NiO nanoparticles, composite PAN/NiO fibers were obtained by electrospinning. The obtained composite fibers were modified with heat treatment (stabilization and carbonization) processes. Obtained C/NiO fibers were investigated by SEM, and EDAX. It was shown that obtained composite fibers could be used for the detection of acetone and acetylene in air. These results show that C/NiO based electrospun fibers have potential applications in gas sensors.

**Keywords:** electrospinning; carbon fibers; NiO nanoparticles; gas sensing

## 1. Introduction

Gas sensors are chemical sensors capable of detecting the presence of certain types of gases [1]. Over the centuries, various gas sensor technologies have been used to detect various gases, including semiconductor gas sensors, catalytic gas sensors, electrochemical gas sensors, optical gas sensors, and acoustic gas sensors [2]. The performance of each sensor is based on several properties including sensitivity, selectivity, limit of detection, response time, and recovery time [3]. The principle of operation of gas sensors is based on the semiconductor properties of the material [1,4]. They operate on the principle of a reversible process of gas adsorption on the surface of a gas-sensitive material [4], which leads to a change in the electrical resistance of the device.

Gas-sensitive materials based on metal oxide have attracted considerable attention for the detection of air pollution as well as accidental leakage of life-threatening flammable, explosive and toxic substances [5,6]. Metal oxide is known to have high sensitivity, short response time and excellent selectivity for various types of gases. Many new technologies have been implemented _to improve and optimize these characteristics [7]. Recent results on the development and fabrication of electroformed gas sensors for the detection of various gases including hydrogen ($H_2$), methane ($CH_4$), nitrogen monoxide (NO), carbon monoxide (CO), hydrogen sulfide ($H_2S$), ammonia ($NH_3$), ethanol ($C_2H_5OH$), acetone ($CH_3COCH_3$), formaldehyde (HCHO) and toluene ($C_7H_8$) are reported [8]. Investigations have shown that nanofibers of different compositions (single phase metal oxides, modified metal oxides, metal oxide nanocomposites and metal oxides combined with carbon nanomaterials) exhibit high response values, long-term stability, low moisture dependence, fast response/recovery time and low gas detection limits [9,10].

Metal oxide nanoparticles are one of the most applicable materials for semiconductor sensors for various applications [11]. The most important parameter for sensors is selectivity, which depends on the microstructure of the sensing material and directly on the manufacturing technology of the sensor electrode. However, sensors obtained using the most common "thin film" technology often have low selectivity and sensitivity when measuring the concentration of gases in the atmosphere due to the diversity of its constituents and morphological characteristics [12].

Metal oxide gas sensing electrodes have received considerable attention due to the surface properties of metal oxides, which is a key consideration for sensors. The surface of metal oxides is a good target for the molecules of the above substances, and also has high adaptability to various gases and efficiency. selectivity due to shorter response time [13,14]. The most common and accessible oxides are nickel oxide nanoparticles, which, in turn, can be studied by simple methods and modified with various components.

In order to produce carbon fiber-based gas sensor, oxide metals such as $ZnO$, $TiO_2$, $CeO_2$, $SnO_2$, $In_2O_3$, $CuO$, $NiO$, $WO_3$ and $Fe_3O_4$ are usually used, depending on the characteristics of the synthesis methods. Table 1 shows the oxides most used in gas sensors. Some synthesis methods and detectable gases are also shown. In turn, metal oxide nanoparticles are produced by reducing the size and separating the bulk materials into small particles by various physical and chemical methods such as sputtering [15], thermal evaporation [16], pulsed laser ablation [17] and mechanical (milling and grinding) [18], chemical vapor deposition (CVD) [19], sol-gel [20,21], hydrothermal synthesis [22] and solution combustion [23].

**Table 1.** Summary of studies of metal oxide nanoparticles for gas sensing.

| Metal Oxide Nanoparticles | Properties of Nanoparticles | Method of Synthesis | Fabrication Technology of Gas Sensors Electrodes | Sensing Gas | Ref. |
|---|---|---|---|---|---|
| $TiO_2$ | Size: 3 nm–30 nm | Sol–gel | Coaxial electrospinning | Ethanol 100 ppm Methanol 100 ppm CO 100–300 ppm $NO_2$ 0.5–4 ppm | [24–26] |
| $SnO_2$ | 17.8 nm | Aerosol flame reactor (Premixed Flame) | Electrospinning | VOC (ethanol, acetone, 20–200 ppm) Triethylamine 50 ppm | [27] |
| $CuO$ | 20–100 nm | Thermal deposition method | Electrospinning | Ethanol 250 ppm, hydrogen 250 ppm and liquefied petroleum gas 2500 ppm | [28] |
| $ZnO$ | - | Wet chemical route | Electrospinning | 1-propanol 15 ppm acetone 15 ppm methanol 15 ppm | [25] |
| $In_2O_3$ | 10–20 nm | Film is grown by molecular beam epitaxy (MBE) | Electrospinning | Carbon monoxide 100 ppm | [29–31] |
| $TiO_2$-$CeO_2$ | Ave grain size: 17–28 nm | Sol–gel | Electrospinning | Carbon monoxide 25–400 ppm | [32] |
| $NiO$ | 95.5–417.2 nm | Solution Combustion [33] | Electrospinning | Acetylene—50 ppm Acetone—50 ppm | This work |

Electrospinning is a versatile and flexible method for obtaining nanofibers. Today, nanofibers are produced from a variety of materials, including natural and synthetic polymers, carbon-based nanomaterials, and composite materials [34]. Currently, the electrospinning method is represented by a wide range of implementation modes: from single-needle (the basic configuration most often used in laboratories) and multi-needle [35] to devices for electrospinning without needles and with a free liquid surface [36]. Currently, there is an increased interest in the process of electrospinning. Recent advances in this field give promising predictions for the application of electrospinning in tissue engineering in combination with rotational bioprinting [37] and 3D printing [38], to create magnetic [39] and ceramic fibers [40].

This work can provide valuable information for researchers in the field of electro-spinning gas sensors based on metal oxides, especially nickel oxide in polymeric carbon fibers. Thus, smart systems based on gas sensors have drastically entered everyday life and have found increasing use in applications related to health and safety [41], such as medical diagnostics [42], atmospheric air quality monitoring [43], food processing [44] and detection of toxic/flammable and explosive gases, and control of limited emissions [45]. These and other applications in harsh industrial environments require the development of sensors that are fast, sensitive, selective, reliable as well as inexpensive. This development aims to produce such portable gas analyzers based on nanomaterials. Carbon, which is the main component of carbon fibers, is not a gas-sensitive material, and carbon/NiO composite fibers have been prepared to give it the desired properties. NiO is a p-type oxide with a band gap of 3.6–4.2 eV, which is widely used to obtain gas-sensitive materials, which is characterized by thermodynamic stability in comparison with other metal oxides [46]. The main task of improving the quality of the use of nickel oxide is to change the structural parameters of the material. For this purpose, a one-dimensional matrix can be formed with the inclusion of particles of nickel oxide. In the present work, a sensor based on polymer containing nickel oxide nanoparticles is proposed. It has been shown that nickel oxide nanoparticles can be obtained by a simple and cheap method; solution combustion. Based on nickel oxide, C/NiO composite fibers were obtained, which are effective in the detection of volatile organic compounds using acetone and acetylene as an example.

## 2. Materials and Methods

### 2.1. Synthesis of Nickel Oxide Nanoparticles by the Solution Combustion Method

In this work, the method of liquid-phase combustion (solution combustion synthesis, SCS) was used to obtain nanodispersed nickel oxide. For the synthesis of nickel oxide nanoparticles by the solution combustion method, nickel nitrate hexahydrate ($Ni(NO_3)_2 \cdot 6H_2O$) was used as an oxidizer and urea ($NH_2CONH_2$) as a fuel. The synthesis of ultrafine metal oxide particles is based on the exothermic process of liquid-phase interaction of system components, including fuel and oxidizer.

The reagents were completely dissolved in a thermostable chemical beaker in distilled water, then evaporated to a volume of 5–7 mL on heating plate C-MAG HP 4 (Germany). After evaporation, the mixture was heated to 260 °C; after evaporation of the main volume of water and transition to a gel-like system, self-ignition of the solution was observed. The auto-ignition temperature is chosen based on the decomposition temperature of urea, at which the decomposition of the fuel occurs with an increase in temperature up to 1200 °C. At this temperature, instant ignition of the mixture occurs, the final product settles directly on the walls and bottom of the beaker. The reaction product is an ultrafine black powder.

### 2.2. Electrospinning Process of PAN/NiO Fibers

Carbon-based fibers were obtained from PAN by electrospinning. In the work polyacry-lonitrile ((-CH$_2$-CH(CN)-)$_n$, molecular weight 150,000 g/mol, manufactured by DFL Minmet Refractories Corp.) and dimethylformamide (DMF, $(CH_3)_2NC(O)H$, Sigma-Aldrich ≥ 99.8%) were used. At the first stage, 9 wt. % PAN/DMF solution was prepared. This concentration is optimal, which was established experimentally. Concentrations from 7 to wt. 11% have been studied. At high concentrations, the solution is too viscous, making it difficult to spin the fibers. At low concentrations, the formation of globular and spherical structures in the fibers is observed. To prepare a PAN/DMF solution, a corresponding weighed portion of powdered PAN was mixed with dimethylformamide. Nickel oxide powder was added in small portions to the resulting PAN/DMF solution in a ratio of 7:3 by weight (PAN:NiO) without taking into account DMF. The resulting suspension was continuously stirred on magnetic stirrer (ISOLAB) for 12 h to uniformly distribute the nanoparticles.

The process of electrospinning of fibers was carried out on an installation (electrospin-ning setup AME-HZ-10) consisting of a solution supply system: a syringe reservoir with a metal needle, a voltage source, and a drum collector with adjustable rotation speed. A

5 mL syringe was filled with a PAN/NiO/DMF suspension; a metal needle was placed at the end of the syringe, to which a negative voltage was applied, and a positive voltage was applied to the collector. The voltage applied to the needle and drum collector was 15 kV, the solution feed rate was 1.0 mL/h, the distance between the needle and the collector was 15 cm, and the surface of the drum collector for fiber deposition was covered with aluminum foil. The applied voltage, the distance between the electrode and the collector were adjusted to obtain a stable electrospinning jet.

### 2.3. Stabilization and Carbonization of PAN/NiO Fibers

The original PAN/NiO electrospun fibers were subjected to stabilization and carbonization processes. The initial PAN/NiO fibers were stabilized in air in the CVD furnace (three heat zone tube furnace, 1200 Protech) with a quartz tube with diameter 6 cm. The reactor was heated to a temperature of 260 °C. The stabilization time was 1 h; after the stabilization process, heating of the reactor was stopped, the sample was cooled to room temperature without being removed from the reactor in an air atmosphere.

The stabilized fibers were carbonized in the same CVD furnace. The quartz tube was preliminarily purged with argon (99.993%) to remove air from the reactor and exclude contact with oxygen. The reactor was heated to a temperature of 700 °C. The carbonization time was 1 h; after the end of the carbonization process, heating of the reactor was stopped, the sample was cooled to room temperature without being removed from the reactor in an argon atmosphere.

### 2.4. Gas Sensing by NiO/C Composite Fibers

To establish the sensitivity to gases of the obtained composite carbon fibers with the addition of nickel oxide particles, tests were carried out. NiO/C fibers were placed in an agate mortar and thoroughly ground, then the ground composite fibers were mixed with polyvinylidene fluoride (PVDF), which was used as a binder in a mass ratio: NiO/C:PVDF—85:15. The prepared viscous suspension was applied to a microchip, which is a microchip of interdigitated gold nanowires with two electrodes fixed on aluminum oxide. After applying the suspension, the sample was dried for 2 h in an oven at a temperature of 100 °C. The finished sample was placed in a cylindrical glass chamber, and the electrodes were connected to a P-45X potentiostat-galvanostat ("Electrochemical Instruments") workstation to measure the current characteristic between these two electrodes when exposed to the analyzed gas and/or vapor. The gas sensitive properties have been verified by static tests. The sensor was placed in a 10 L test chamber Low concentration gaseous acetylene or acetone is produced by proportionately mixing 50 ppm bottled acetylene or acetone gas with bottled nitrogen and oxygen (special purity). The flow of gases is controlled by gas mixers Alicat (Scientific M series). The total gas flow into the chamber was 500 sccm. A volume of steam saturated with acetone was injected into the chamber to create an acetone environment at a certain concentration, and the gas detection characteristics of the sensor were recorded in the PC software. All changes in the resistance of the sensor were recorded in real time. For stabilization, the gas sensor was previously placed in the test chamber for 1 h in the prepared gaseous medium. The sensor response was determined as $R = (I_a - I_g)/I_g$ ($I_a$ is the sensor current in air, $I_g$ is the sensor current in acetone). A total of 10 test runs of the NiO-based gas sensor were carried out. The control experiment has the closest value to the average values for all experiments.

### 2.5. Methods of Characterization

The investigation of the structure and morphology of the obtained samples was carried out on a Quanta 200i 3D scanning electron microscope (FEI Company, USA) in normal secondary electron (SE) mode equipped with an energy dispersive X-ray analysis (EDAX) system with an accelerating voltage of 30 kV (National Nanotechnology Laboratory of Open Type (NNLOT), Al-Farabi KazNU, Almaty, Kazakhstan). Scanning electron microscopy imaging in secondary electron mode was carried out. The study of EDAX was carried out

on 5 points of the sample and the standard deviation was derived. X-ray diffraction (XRD) analysis was performed on a Dron-4 diffractometer (Institute of Combustion Problems, Almaty, Kazakhstan). To determine the specific surface area, there was a single-point BET method based on physical low-temperature gas adsorption. Nickel oxide samples were studied on a Sorbtometer-M analyzer. A more detailed description of the methods for studying the porous materials is given in research work about SPS technique [47].

## 3. Results

### 3.1. Physicochemical Properties of Nickel Oxide Nanoparticles

The XRD patterns of the NiO nanoparticles showed diffraction peaks absorbed at 2θ values (Figure 1a). SEM results and the distribution of the diameter of nickel oxide nanoparticles also are presented on the Figure 1.

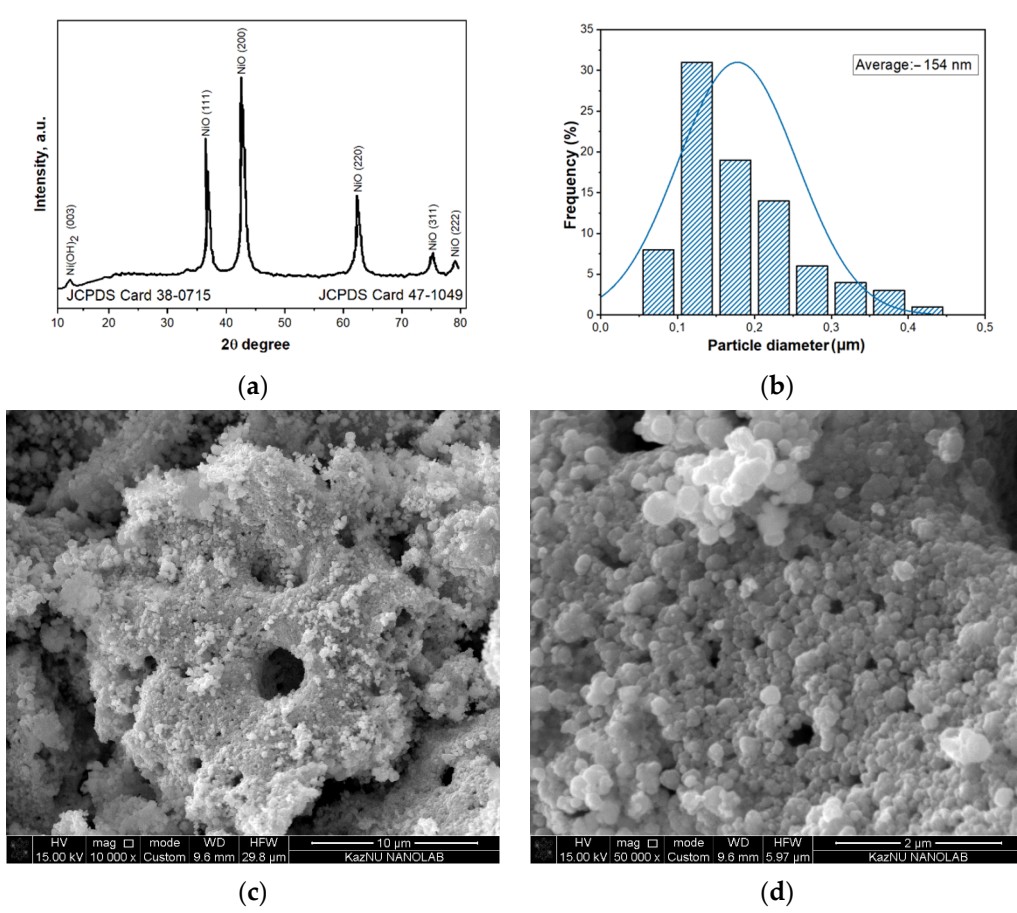

**Figure 1.** Results of characterization nickel oxide nanoparticles: (**a**) XRD, (**b**) distribution of the diameter of NiO nanoparticles, (**c**,**d**) SEM in SE mode.

XRD diffraction pattern includes 5 peaks. The diffraction peaks at 37.27, 43.27, 62.84, 75.42 and 79.36° of 2θ correspond cubic NiO crystallites with various diffracting planes (111), (200), (220), (311) and (222). All these diffraction peaks accordance with the standard spectrum (JCPDS Card 47-1049). The small peak at 12° most likely corresponds to small impurities of $Ni(OH)_2$, however, other peaks corresponding to this compound were not identified, in particular due to the superposition of more intense peaks. According to the obtained XRD data, it can be noted that the sample is a single-phase nickel oxide. The nickel oxide crystallite calculate size using the Scherrer equation is 480 Å. As can be seen from the results of SEM analysis, the sizes of nickel oxide nanoparticles lie in the range from 95.5 to 417.2 nm with an average size of 154 nm, have a spherical structure, and the formation of agglomerates is also observed. The agglomeration may occur due to the crystallites being of nanodimension and a large number of uncompensated surface bonds. The nanocrystals

possess large surface energy, which leads the nanocrystals to aggregate in order to lower their surface energy during synthesis. Crystallinity is evaluated through comparison of crystallite size determined using SEM and XRD. The crystallinity index was calculated according to the equation:

$$I = \frac{D_p(\text{SEM})}{D_{cry}(\text{XRD})}$$

In our case, the crystallinity index is 3.2, which indicates a polycrystalline state of the substance (single-crystalline materials have a crystallinity index close to 1) [48]. The specific surface area for NiO nanoparticles is 65.544 m$^2$/g, which is a good indicator for oxide materials. The authors [49] synthesized nickel oxide by sol-gel method with a specific surface area of 5.8 m$^2$/g. A comparison of the specific surface area indicates that the nickel oxide obtained by the solution combustion method has a lower dimensionality and a high porosity.

### 3.2. Morphology of NiO/C Composite Fibers

PAN/NiO fibers were obtained by electrospinning, then they were subjected to stabilization and carbonization processes. Figure 2 shows the SEM image and EDAX spectrum of PAN/NiO composite fibers after stabilization and carbonization processes, respectively. As can be seen from the results of the analysis, the PAN/NiO composite fibers after the stabilization process and subsequent carbonization at a temperature of 700 °C retain a stable one-dimensional structure with a diameter from 150 nm to 300 nm (av. 291 nm). Numerous crystalline inclusions of NiO are present. NiO particles have better adhesion to the fiber surface, because during the carbonization process, chemical and physical changes occur that improve the adhesion between NiO nanoparticles and the fiber surface. The ratio of the main elements of the fibers changes; after the carbonization process, the carbon content is 81 wt. %, the nickel content is 16 wt. %. The increased nickel content is explained by the formation of volatile carbon- containing compounds, which leads to a decrease in the total carbon content and, consequently, an increase in the mass fraction of nickel oxide.

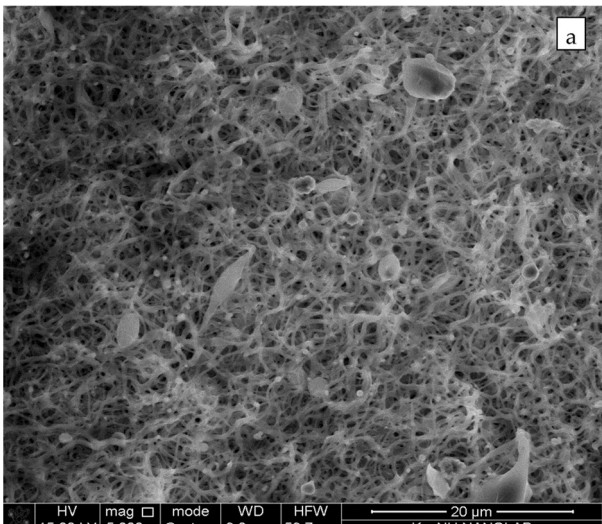 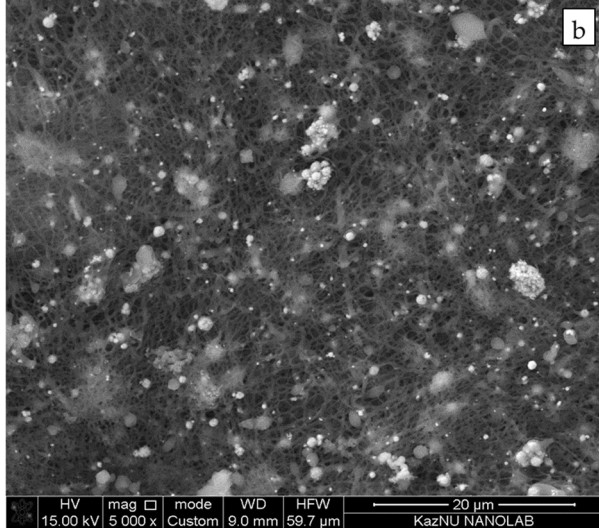

**Figure 2.** *Cont.*

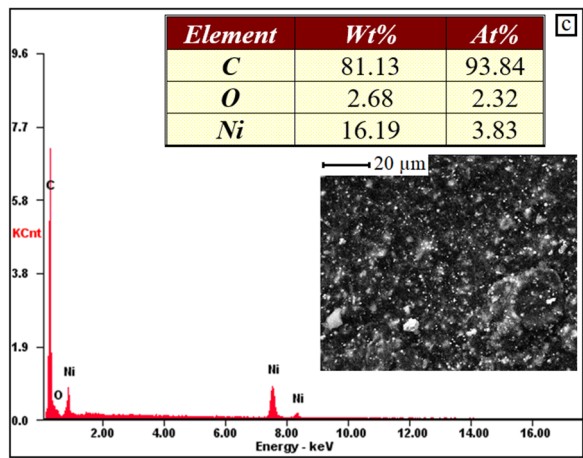
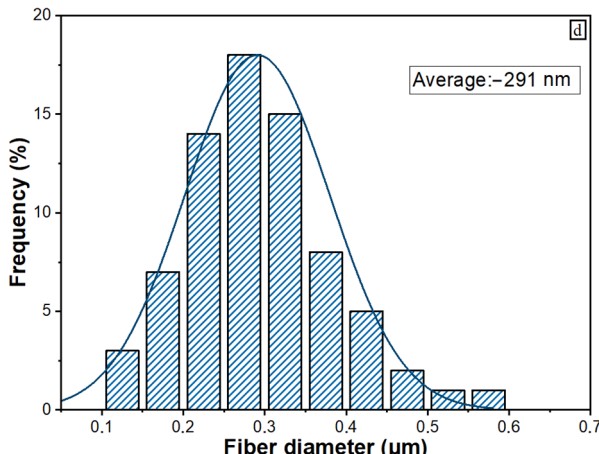

**Figure 2.** Results of characterization: (**a**) SEM in SE mode image of intermediate PAN/NiO fibers after stabilization at 260 °C, (**b**) SEM in SE mode image of NiO/C fibers from PAN/NiO after stabilization at 260 °C and carbonization at 700 °C; (**c**) EDAX spectrum of NiO/C fibers after carbonization and (**d**) NiO/C fiber diameter distribution after carbonization.

### 3.3. Gas Sensitive Characteristics of NiO/C Composite Fibers

The electrical response of the gas sensitive material to volatile organic compounds (acetylene and acetone) was tested. When exposed to acetone and acetylene, a change in resistivity is clearly observed, which is due to the sensitivity of the material to these gases. Figure 3a,b shows graphs of the conductivity of NiO/C fibers as a function of time.

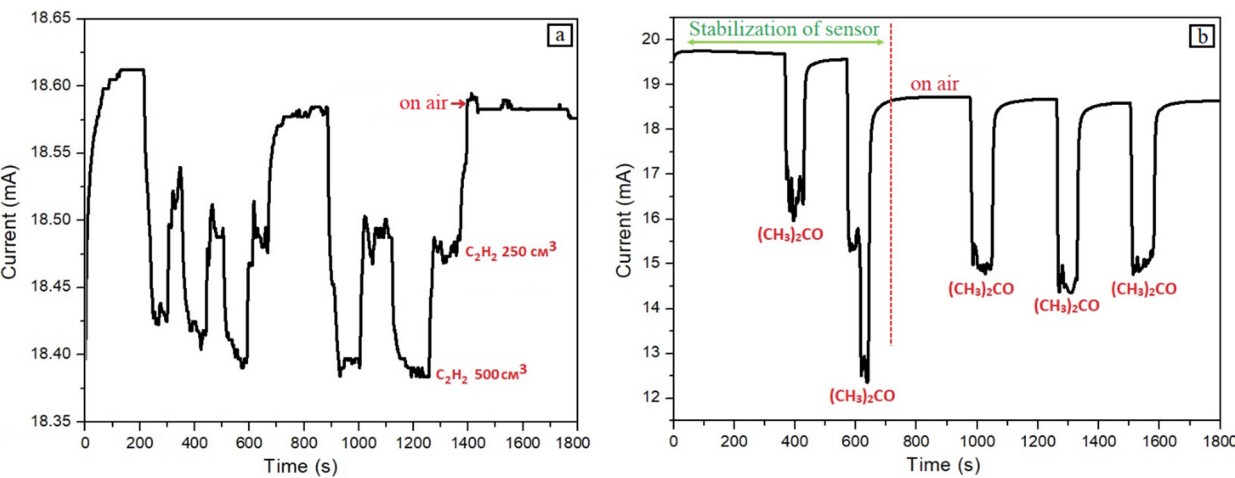

**Figure 3.** Graphs of the conductivity of NiO/C fibers from time for a mixture of: (**a**) acetylene/air and (**b**) acetone/air.

As can be seen from the graph in Figure 3a, a change in resistance depends on the concentration of the analyzed gas. When exposed to adsorbate gas, an increase in resistance was found. After purging with air, the material regains its resistance. The sensing mechanism of NiO-based gas sensors involves three interactions: adsorption-oxidation-desorption. When the sensor is in the air, oxygen molecules react with NiO surface. Electrons on the NiO surface combine with $O_2$ to form oxygen negative ions. This process causes a decrease in electrons and the increase in holes to form a hole accumulation layer, resulting in the resistance of the sensor decreases correspondingly. When NiO surface comes into contact with $C_2H_2$ gas, oxygen ions will oxidize gas molecules into $CO_2$ and $H_2O$, and it releases electrons to recombine with holes, leading to the decrease in carriers in hole accumulation layer and an increase in the resistance. A more detailed description of the gas detection mechanism for NiO-based sensors can be found in research work about $C_2H_2$

gas sensing [50]. Further developments of the work will be connected with the study of the selectivity of the obtained sensors, as well as the study of detection in relation to other gases.

## 4. Conclusions

Gas-sensitive materials based on carbon fibers with the addition of nickel oxide nanoparticles were obtained. Nickel oxide nanoparticles were synthesized by a simple and inexpensive solution combustion method. The sizes of nickel oxide nanoparticles lie in the range from 95.5 to 417.2 nm with an average size of 154 nm and have a spherical structure. NiO/C fibers were obtained by electrospinning a suspension of PAN/NiO, with further stabilization at 260 °C in air and carbonization at 700 °C in argon. NiO/C fibers are one-dimensional structures with a diameter from 150 nm to 300 nm (av. 291 nm). According to the EDAX data, the fibers consist of carbon ~81%, nickel ~16% and oxygen ~3% by wt. The fibers have bead-like inclusions of crystalline nickel oxide. NiO/C fibers were tested as a gas sensitive material. The gas sensitive characteristics of NiO/C fibers were studied in relation to the detection of acetylene and acetone vapor. Analysis of the gas sensitivity of the NiO/C fibers showed good stability and high sensitivity to the acetone vapor.

**Author Contributions:** B.K.: Conceptualization, Writing–original draft, investigation. G.S.: Writing–review and editing, visualization. A.I.: Investigation, writing–review and editing. Z.M.: Supervision, Writing–original draft, Methodology. All authors have read and agreed to the published version of the manuscript.

**Funding:** This research was funded by the Committee of Science of the Ministry of Education and Science of the Republic of Kazakhstan, grant number and title: "OR11465430 Development of new composite and structural materials for the development of the innovative industry of the Republic of Kazakhstan" under the subprogram "Development of technology for obtaining carbon fibers and their application as sensors and carbon composite material". The APC was funded by "OR11465430".

**Data Availability Statement:** Data are available upon request.

**Acknowledgments:** This work was acknowledging the Committee of Science of the Ministry of Education and Science of the Republic of Kazakhstan; Project of "Development of technology for obtaining carbon fibers and their application as sensors and carbon composite material".

**Conflicts of Interest:** The authors declare no conflict of interest.

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
