# Peer review of "Gas Sensitive Materials Based on Polyacrylonitrile Fibers and Nickel Oxide Nanoparticles"

_jcs, doi:10.3390/jcs6110326_

Round 1

Author Response

The Authors are grateful to the reviewer for the work done. The comments made have significantly improved the quality of the paper. And the presentation became much clearer.

Reviewer 2 Report

The work is devoted to the synthesis of polyacrylonitrile-NiO composite fibers by electrospinning. Nickel oxide nanoparticles were synthesized by solution combustion synthesis from nickel nitrate hexahydrate. The results show that electrospun C/NiO-based fibers have potential applications in gas sensors.

The work deserves publication in J. Compos. Sci. but with the following significant revisions.

1. Introduction

-It mainly discusses the application of sensors in the gas medium of various organic molecules, but there is no method of synthesis of these catalysts how they are obtained. As I understand the method stated by the authors is a combination of methods of combustion + templat synthesis in the form of polyacrylonitrile fibers. I ask the author to reveal questions about the synthesis of such "wet" materials (sol-gel (10.1007/s10971-013-3039-0, 10.1016/j.powtec.2020.04.040), hydrothermal, microwave). Also interested in the subject as a study of porous materials 10.1016/j.matchar.2018.08.044.

2. MATERIALS AND METHODS

-The work uses modern methods of research, the results obtained can be trusted

-The paper studied the porosity of materials using BET - but there is no information what is the principle of the method of the device... Low temperature gas adsorption or other? No hysteresis loop indicated, for understanding and types of pores.

Results

-Figure 1. Specify after which powders were studied (judging from the description it is nickel nanoparticles) please add in the caption of the figures.

-I would like to compare the obtained XRD spectra with the given database, please indicate the number of the card from the international database given in PDF-2. and compare this data with the obtained powder

-No specific description of the results obtained. It feels like the authors lost a piece of text when describing the main functional part of the study material. How many cycles the material was run, how selective to other molecules. no description. please expand

-Lack of comparison on how much qualitative received catalysts? please enter in the table with characteristics (surface, sensor on molecules, method of synthesis, composition, etc.) about similar catalysts on the basis of metal oxides.

Conclusions

-Please add in the conclusions results describing the material itself, not what methods it was described. Surface, composition, sensory abilities...

Author Response

(The authors gave the same response as above.)

Round 2

Author Response

(The authors gave the same response as above.)

Reviewer 2 Report

To accept the article as it stands

Author Response

(The authors gave the same response as above.)

Round 3

Reviewer 1 Report

The authors did not remove the sentence "The standard deviation for the EDAX study was ±3". It cannot be a correct information. Even if it is anyhow, it is not mentioned for which point, and for how many data this SD was calculated, which will leave the readers with confusion. This sentence needs to be removed or the correct data needs to be implemented (which is beyond the author's ability).